# ALLoRA: Adaptive Learning Rate Mitigates LoRA Fatal Flaws

## Abstract

Low-Rank Adaptation (LoRA) is the bread and butter of Large Language Model (LLM) finetuning. LoRA learns an additive low-rank perturbation of a pretrained matrix parameter to align the model to a new task or dataset. We identify three core limitations to LoRA for finetuning–with only a limited amount of training steps. First, it employs Dropout as a means to prevent overfitting. We prove that Dropout is only suitable for long training episodes but fails to reliably regularize training for short training episodes, e.g., finetuning. Second, LoRA's parameters initialization is at 0 makes the optimization landscape poorly conditioned during the first steps of training. That poor conditioning combined with the need to move away from 0 lead to slow training dynamics. Third, the scaling factor that multiply each LoRA additive perturbation create "short-sighted" interactions between the LoRA modules of different layers. Motivated by principled analysis of those limitations, we find an elegant solution: a Dropout-free, scaling-free, LoRA with Adaptive Learning rate–coined ALLoRA. By scaling the per sample and per parameter gradients with a coefficient inversely proportional to parameters' $\ell_2$ norm, ALLoRA alleviates those three limitations. As a by-product, ALLoRA removes two hyper-parameters from LoRA: the scaling factor and the dropout rate. Empirical results show that ALLoRA admits better accuracy than LoRA on various settings, including against recent LoRA variants such as Weight-Decomposed Low-Rank Adaptation (DoRA). Ablation studies show our solution is the optimal in a family of weight-dependent / output-dependent approaches.

## 1 Introduction

Large Language Models (LLMs) Hoffmann et al. (2022); Touvron et al. (2023); Jiang et al. (2023) are Deep Neural Networks (DNNs)–commonly built from Transformer with self-attention–built for sequence processing, e.g., Natural Language Processing (NLP). LLMs have radically changed the way we approach NLP Chowdhary (2020) by removing the need for handcrafted feature engineering such as bags of words Zhang et al. (2010). Instead, current solutions directly operate on the input data–or a lossless compression known as tokens Shibata et al. (1999). Because we now have access to humongous amount of text data, the standard training pipeline for LLMs take the following form. First, the LLM is *pretrained* a large text corpus through next-token prediction. That autoregressive pretext-task enables the LLM to learn the underlying dynamic of the language. Commonly, RLHF is also employed after pretraining to make the model's behavior shift from autoregressive to agentic. Then, the LLM is *fine-tuned* on a more specific downstream task or dataset. That fine-tuning is user-specific and plays a fundamental role in making LLMs practically useful but relies on much more limited datasets, as we formalize below.

> **Premise:** *The training regime involved in pretraining and fine-tuning are drastically different. The former takes places on large "industrial" computational clusters with limitless data and training steps. The latter takes place on small user-owned computational resources with limited data and training steps.*

That premise is now widely accepted upon as the latest state-of-the-art LLM solutions stem from the numerous open-source industry groups such as Meta's Llama Dubey et al. (2024), Google's Gemma Team et al. (2024), Apple's OpenELM Mehta et al. (2024), or Cohere's Aya. Hence, as

LLM practitioners, most of the attention is now turning into deriving fine-tuning strategies that meet the very particular needs of fine-tuning LLMs.

To tackle that paradigm shift introduced by the pretraining-finetuning strategy, specialized methods have been developed, such as the eponymous Low-Rank Adaption (LoRA). LoRA has fueled countless deployment of LLMs–as it took a gigantic leap in accommodating for the fine-tuning regime. In short, LoRA proposes to fine-tune a LLM by learning an additive low-rank matrix perturbation to some of the LLM's internal parameter matrices. Core to its design, LoRA leverages (i) Dropout Srivastava et al. (2014) as a mean to prevent overfitting to the fine-tuning task, and (ii) zero-initialization to ensure that training starts from the LLM's pretrained mapping, and (iii) a scaling factor that rescales the LoRA's matrix factorization. While the impact of LoRA is ubiquitous, we nonetheless believe that LoRA could be further improved based on three observations.

> ***LoRA's three fatal flaws for finetuning:*** *First,* **Dropout**–*a stochastic regularizer–whose benefits quickly vanish when considering fine-tuning, and can in fact introduce detrimental additional variance to the training. Second, the* **zero-initialization** *which is difficult to escape from as the fine-tuning parameters start from a saddle point. Third, the* **scaling parameter** *that introduces nonlinear interactions between LoRA modules of different layers.*

While each of those three design choices are well-motivated when considering long training, e.g., pretraining, it becomes harder to prove their benefits when considering fine-tuning that only employs a minimal amount of training steps. That is why, after carefully bringing to light and studying the above flaws of LoRA–in the context of fine-tuning–in section 3, we will propose a novel variation of LoRA that we coin **ALLoRA** for Adaptive Learning rate LoRA (section 4). ALLoRA proposes to remove the Dropout regularizer and the scaling factor while adding an adaptive learning rate for the low-rank matrices entries. As depicted in listing 1, the implementation is straightforward with theoretical and practical benefits. First, by removing the Dropout regularization and the scaling factor, ALLoRA is simpler to employ as it no longer requires cross-validation of those parameters. Second, we demonstrate how ALLoRA improves performances over LoRA and alternatives such as DoRA. In short, our adaptive learning rate strategy is able to prevent over-fitting, learn competitive solutions, and converge more quickly than alternatives–all while employing less hyper-parameters.

We summarize our contributions below:

1. We identify three inefficiencies (sections 3.1 to 3.3) in the current LoRA design that make it unfit for short training, i.e., finetuning.

2. We propose a novel adaptive learning rate variation of LoRA–coined ALLoRA–free of two of the original LoRA's complicated designs: the Dropout regularizer and the scaling factor.

3. We empirically validate the benefits of ALLoRA over numerous datasets and model architectures including the latest Llama3 family. We obtain that despite ALLoRA employing less hyperparameters than LoRA, it is able to outperform its counter part and recent variants such as DoRA consistently.

The full codebase to reproduce figures and tables will be provided upon completion of the review process.

## 2 RELATED WORKS

LoRA is a type of *Parameter Efficient Fine Tuning* (PEFT) method designed to reduce the cost of finetuning LLMs. As LLMs typically have large number of parameters–in the scale of billions–one can not afford to finetune all those parameters on a particular downstream task or dataset. Existing PEFT can be divided into three categories, namely *Adapter-based Methods*, *Prompt-based Methods*, and *LoRA*.

Adapter-based methods (Houlsby et al. (2019), He et al. (2022), Karimi Mahabadi et al. (2021b), and Karimi Mahabadi et al. (2021a)) introduce additional trainable modules, *a.k.a.* the *adapters*, into the original backbone whose weights are frozen during the finetuning. In Houlsby et al. (2019), linear modules were added in sequence to the existing layer, while in He et al. (2022), they were added in parallel to the existing layer for the sake of better performance.

Prompt-based methods (Lester et al. (2021), Razdaibiedina et al. (2023), and Wang et al. (2023)) introduce soft tokens as trainable parameters and prepend them to the prompt. This category is the least intrusive as the finetuning can be done by only prompting the LLMs. However, prompt-based methods are in general sensitive to initialization and their overall effectiveness is affected.

LoRA Hu et al. (2021) uses low-rank matrices to simulate weight changes of the pretrained weights. Since low-rank matrices can be merged back to original weights, LoRA does not incur any additional cost at inference, which is a significant advantage over the other two categories. Many variants were proposed lately. For example, in Zhang et al. (2023), SVD decomposition was employed to determine significance of singular values, and less important ones are pruned. Hyeon-Woo et al. (2022) applies low-rank Hadamard product to federated learning. Qiu et al. (2023) and Liu et al. (2024b) adopt orthogonal factorization and applied to diffusion models. Renduchintala et al. (2023) introduces weight tying and realizes more savings on number of parameters. A unified LoRA family was introduced for Stable diffusion in Yeh et al. (2024). Different combinations of LoRA are chosen for different tasks in Ponti et al. (2022). A scaling vectors is learnt to adjust a pair of frozen random matrices shared across layers in Kopiczko et al. (2023).

More recently, Liu et al. (2024a) proposes decomposing the weights into directional and magnitude components to boost accuracy. Hayou et al. (2024a) studies the optimal initialization of the low-rank matrices, and a follow-up work Hayou et al. (2024b) proposes to apply different learning rate to different low-rank matrices. Superficially this is similar to one of our idea to adapt learning rate, though our idea is inspired by a principled study of dropout (Srivastava et al. (2014)).

More broadly, Zhao et al. (2024) applies the low-rank concept to compute low-rank gradients directly. Jang et al. (2024) provides rigours theatrical study on the existence and convergence of LoRA solutions. And Zhang & Pilanci (2024) is a study of the potential ill conditioned low-rank matrices.

## 3 A CRITICAL ANALYSIS OF LoRA FOR FINETUNING

Because PEFT is the current bottleneck between large and powerful LLMs and specialized practical use-cases, numerous variations of LoRA have emerged.

**Definition 1.** (Low Rank Adapters (LoRA) from Hu et al. (2021)). For any weight matrix $\boldsymbol{W} \in \mathbb{R}^{n_1 \times n_2}$ in the pretrained model, we constrain its update in the finetuning process by representing the latter with a low-rank decomposition $\boldsymbol{W} = \boldsymbol{W}^* + \frac{\alpha}{r}\boldsymbol{BA}$. Here, only the weight matrices $\boldsymbol{B} \in \mathbb{R}^{n_1 \times r}$, $\boldsymbol{A} \in \mathbb{R}^{r \times n_2}$ are trainable. The rank $r \ll \min(n_1, n_2)$ and $\alpha \in \mathbb{R}$ are tunable constants.

Yet, very little attention was put on the core premises of LoRA within a finetuning context, i.e., with limited amount of training steps. We propose here a principled and critical study of LoRA, culminating with our finding of three core flaws of LoRA for short training (sections 3.1 to 3.3). The following section 4 will investigate our solution, ALLoRA.

### 3.1 FIRST FLAW: A STOCHASTIC REGULARIZATION THAT WILL NOT CONVERGE

LoRA's regularization hinges on using Dropout Srivastava et al. (2014), i.e., on applying a multiplicative binary mask to the latent space of the matrix factorization. In fact, it has been long known that Dropout is a great solution to prevent overfitting. However, those benefits have either been found in the context of (pre-)training, or through theoretical derivations assuming infinite training time, i.e., *on expectation*. For example, Wager et al. (2013) shows how in the linear regime dropout acts as some variant of $\ell_2$ regularization on normalized design matrix (inputs), a result also found in the previous study of Wang & Manning (2013). However, we now argue that both previous empirical and theoretical studies showing the benefits of dropout *do not hold in a fine-tuning setting where the number of training steps is highly limited*.

Let's consider in the linear regime the following Ordinary Least Squares (OLS) setting $\|\boldsymbol{Y} - (\boldsymbol{XW}) \odot \boldsymbol{V}\|_F^2$, with $\boldsymbol{Y} \in \mathbb{R}^{N \times C}$, $\boldsymbol{X} \in \mathbb{R}^{N \times D}$, $\boldsymbol{W} \in \mathbb{R}^{D \times C}$ and the random realization of dropout matrix $\boldsymbol{V} \in \{0, \frac{1}{p}\}^{N \times C}$. We note that such parametrization of Dropout is commonly employed in the literature to ensure that its expectation is equal to 1. We have the following property that has

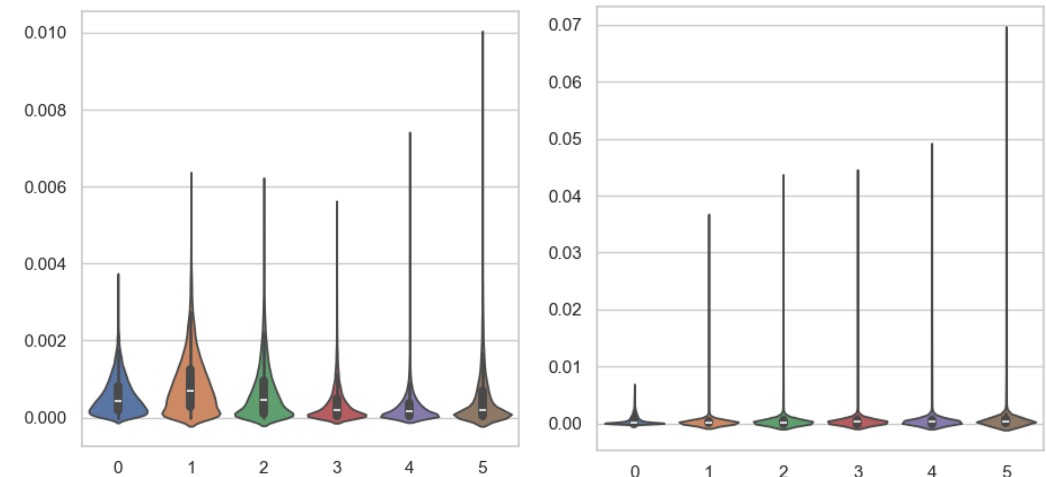

Figure 1: Depiction of the distribution of standard deviation of gradients (y-axis) w.r.t. the second layer of a MLP trained for MNIST (left) and CIFAR10 (right) classification, equipped with Dropout. At each training epoch (x-axis), we consider a single mini-batch and compute the gradients under numerous Dropout realisation. For each entry in the matrix of gradients, we compute the standard deviation and report the distribution over entries. We clearly see that **while the average variance of the gradient decreases slightly during training, the worst case increases, hence leading to unstable training in finetuning regimes**.

motivated the use of Dropout through more than a decade by now:

$$\min_{\boldsymbol{W}} \mathbb{E}\left[\|\boldsymbol{Y} - (\boldsymbol{XW}) \odot \boldsymbol{V}\|_F^2\right]$$

$$= \min_{\boldsymbol{W}} \|\boldsymbol{Y}\|_F^2 + \mathbb{E}\left[-2Tr\left(\boldsymbol{Y}\left(\boldsymbol{V}^\top \odot (\boldsymbol{XW})^\top\right)\right) + Tr\left(\left(\boldsymbol{V}^\top \odot (\boldsymbol{XW})^\top\right)((\boldsymbol{XW}) \odot \boldsymbol{V})\right)\right]$$

$$= \min_{\boldsymbol{W}} \|\boldsymbol{Y}\|_F^2 - 2Tr\left(\boldsymbol{Y}(\boldsymbol{XW})^\top\right) + Tr\left((\boldsymbol{XW})^\top(\boldsymbol{XW})\right)/p,$$

which is solved for $\boldsymbol{W} = p(\boldsymbol{X}^\top\boldsymbol{X})^{-1}\boldsymbol{Y}^\top\boldsymbol{X}$. Comparing that with the usual Tikhonov regularization that produces $\boldsymbol{W} = (\boldsymbol{X}^\top\boldsymbol{X} + \lambda\boldsymbol{I})^{-1}\boldsymbol{Y}^\top\boldsymbol{X}$ we see than whenever the eigenvalues of $\boldsymbol{X}^\top\boldsymbol{X}$ are all identical to a positive constant $c$, e.g. when $\boldsymbol{X}$ is whitened, then $\boldsymbol{W} = \frac{c}{c+\lambda}(\boldsymbol{X}^\top\boldsymbol{X})^{-1}\boldsymbol{Y}^\top\boldsymbol{X}$ hence recovering the dropout solution. Hence we obtain that (assuming $c = 1$ without loss of generality) $p = \frac{1}{1+\lambda}$. The above results recovers known theoretical analysis of Dropout–showing its benefits as an implicit regularizer. However, those derivations only emerge from taking expectation of the loss, i.e., considering infinite training steps. For us, the question thus turns into the following: *what are the benefits of Dropout as a regularizer for very short training regime such as finetuning?*

A first naive bound on "how far off" is the expectation can be obtained (derivations in appendix A) as follows

$$\mathbb{E}\left[\left|\frac{1}{N}\sum_{n=1}^{N}\|\boldsymbol{Y} - (\boldsymbol{XW}) \odot \boldsymbol{V}_n\|_F^2 - \mathbb{E}\left[\|\boldsymbol{Y} - (\boldsymbol{XW}) \odot \boldsymbol{V}\|_F^2\right]\right|\right] \le \frac{Std\left[\|\boldsymbol{Y} - (\boldsymbol{XW}) \odot \boldsymbol{V}\|_F^2\right]}{\sqrt{N}}.$$

As a result, the benefit of $N$, in our case the number of training steps, only appears for large $N$. This is particularly true as training progresses where the variance of the current approximation against the targets ($Std\left[\|\boldsymbol{Y} - (\boldsymbol{XW}) \odot \boldsymbol{V}\|_F^2\right]$) will decrease on average but increase in the worst case compared to initialization with random matrices, or zero matrices with LoRA. We illustrate that dynamics in Figure 1 looking at the standard deviation of the gradients as a function of Dropout realisation in a simple MLP with MNIST classification task as training progresses. Moving to LLMs, we also confirm that simplified model's intuition with LLM experiments below.

**Empirical validation of the harmful impact of Dropout for short fine-tuning** Empirical results support above theory. We run experiments with **Qwen2-0.5B** on **bias_in_bios**, a classification task to predict the job of an employee given the job description, for various dropout rates and up to

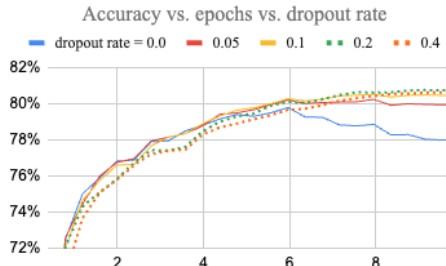 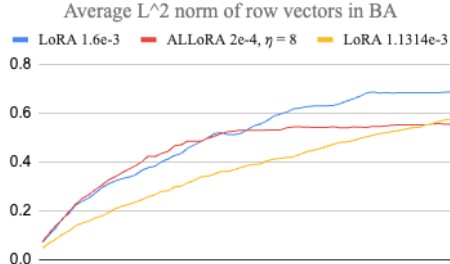

Figure 2: **Left**: LoRA with varying Dropout rates: **High value of Dropout provides the strongest performance after long fine-tuning and the weakest performance after short fine-tuning**. Each line is an average of 3 runs. X-axis is epochs, and Y-axis is accuracy. **Right**: ALLoRA escapes from 0 rapidly, and then tapers off into a measured move. The starting phase matches that of LoRA with a much higher learning rate. LoRA with a lower learning rate can reach the same level of $L^2$ norm but much slower.

10 epochs. We note that 10 epochs is already a large number of finetuning iterations for practical scenarios. It shows that large dropout rates successfully avoid overfitting, but at the cost of lower accuracy at lower number of epochs, while no dropout sees higher accuracy at lower number of epochs, but will overfit at later epochs. See Figure 2 Left and Table 2.

## 3.2 SECOND FLAW: A POOR OPTIMIZATION LANDSCAPE

The second flaw we uncover is the zero-initialization. Let's consider again our simplified model, and applying a LoRA module for finetuning $\mathcal{L}(\boldsymbol{V}) = \|\boldsymbol{X}(\boldsymbol{W} + \boldsymbol{V}) - \boldsymbol{Y}\|_F^2$, where $\boldsymbol{V}$ is our LoRA module, i.e., it is internally a low-rank matrix. We can obtain the following gradient and Hessian of the loss with respect to $\boldsymbol{V}$ as follows. We note that for the Hessian we consider the flattened version of the matrix $\boldsymbol{V}$

$$\nabla_{\boldsymbol{V}}\mathcal{L}(\boldsymbol{V}) = 2\boldsymbol{X}^\top \boldsymbol{X}\boldsymbol{V} - 2(\boldsymbol{Y} - \boldsymbol{X}\boldsymbol{W})^\top \boldsymbol{X}, \tag{1}$$

$$H_{vec(\boldsymbol{V})}\mathcal{L}(\boldsymbol{V}) = (\boldsymbol{I}_n \otimes \boldsymbol{X})^\top (\boldsymbol{I}_n \otimes \boldsymbol{X}) = \boldsymbol{I}_n \otimes \boldsymbol{X}^\top \boldsymbol{X} \tag{2}$$

where the last line comes from the fact that $\|\boldsymbol{X}(\boldsymbol{W} + \boldsymbol{V}) - \boldsymbol{Y}\|_F^2 = \|(\boldsymbol{I}_n \otimes \boldsymbol{X})vec(\boldsymbol{W} + \boldsymbol{V}) - vec(\boldsymbol{Y})\|_2^2$ with $\otimes$ the Kronecker product, and $vec$ the vectorization operator. The Hessian is semi-definite positive, as expected given the convex quadratic nature of the optimization problem. However, as the training set size ($n$) decreases, the optimization landscape for $\boldsymbol{V}$ will become degenerate.

## 3.3 THIRD FLAW: RIPPLE EFFECT OF SCALING FACTOR

Following Definition 1, denote $\eta = \frac{\alpha}{r}$ to be the *Scaling Factor*. The scaling factor plays an important role to match $\|\boldsymbol{B}\boldsymbol{A}\|$ to a comparable level with $\|\boldsymbol{W}^*\|$. Despite its effectiveness, the scaling factor creates a ripple effect across layers of a LLM and may make finetuning unstable. Hu et al. (2021) discussed the importance of the scaling factor and suggest to tune it carefully to prevent $\boldsymbol{B}\boldsymbol{A}$ from overwhelming $\boldsymbol{W}^*$. From a different perspective, Houlsby et al. (2019) empirically showed the scale of the initialization of $\boldsymbol{B}\boldsymbol{A}$ can negatively impact validation accuracy. Later, Hayou et al. (2024a) argued that for the best performance, either $\boldsymbol{A}$ or $\boldsymbol{B}$ must be initialized at 0.

To illustrate the ripple effect, we adopt a multi-linear model which is a simplified version of the toy model in Hayou et al. (2024b).

$$f_l(\boldsymbol{x}) = \boldsymbol{W}_l^* f_{l-1}(\boldsymbol{x}), \quad l \in [L] \tag{3}$$

where $L \geq 1$ is the number of layers. Applying LoRA at each layer gives

$$f_l(\boldsymbol{x}) = (\boldsymbol{W}_l^* + \eta \boldsymbol{B}_l \boldsymbol{A}_l) f_{l-1}(\boldsymbol{x})$$

Expanding the equation, we have

$$f_L(\boldsymbol{x}) = (\boldsymbol{W}_L^* + \eta \boldsymbol{B}_L \boldsymbol{A}_L)...(\boldsymbol{W}_1^* + \eta \boldsymbol{B}_1 \boldsymbol{A}_1)\boldsymbol{x}$$

Let $|| \cdot ||_M$ be a matrix norm induced by a vector norm $|| \cdot ||_v$, we have

$$||f_L(\boldsymbol{x})||_v = ||(\boldsymbol{W}_L^* + \eta \boldsymbol{B}_L \boldsymbol{A}_L)...(\boldsymbol{W}_1^* + \eta \boldsymbol{B}_1 \boldsymbol{A}_1)||_M \cdot ||\boldsymbol{x}||_v$$

We abuse the notation to use $|| \cdot ||$ for both matrix norm and vector norm, applying triangle inequality, we have

$$||f_L(\boldsymbol{x})|| \leq (||\boldsymbol{W}_L^*|| + \eta ||\boldsymbol{B}_L \boldsymbol{A}_L||)...(||\boldsymbol{W}_1^*|| + \eta ||\boldsymbol{B}_1 \boldsymbol{A}_1||)||\boldsymbol{x}||$$

$$= C(1 + \eta \frac{||\boldsymbol{B}_L \boldsymbol{A}_L||}{||\boldsymbol{W}_L^*||})...(1 + \eta \frac{||\boldsymbol{B}_1 \boldsymbol{A}_1||}{||\boldsymbol{W}_1^*||})||\boldsymbol{x}||$$

$$\leq C(1 + \eta \bar{m})^L ||\boldsymbol{x}|| = \Theta((1 + \eta)^L)$$

where $C = ||\boldsymbol{W}_L^*||...||\boldsymbol{W}_1^*||$ is a constant, and $\bar{m} = \frac{1}{L} \sum_{l \in L} \frac{||\boldsymbol{B}_l \boldsymbol{A}_l||}{||\boldsymbol{W}_L^*||}$ is also a constant in a single forward pass. Notice that all the inequalities are tight, we have

**Proposition 1.** (Ripple Effect) In the worst case, a constant scaling factor $\eta$ may cause the final output of a single forward pass of a LoRA finetuned model to grow exponentially *w.r.t.* the number of layers in the model.

We also note that proposition 1 is especially limiting for LLMs that most commonly resort to increased depth rather than increased width to scale up their capacity Hestness et al. (2017). Having concluded our brief tour of LoRA's possible shortcomings when it comes to fine-tuning LLMs in a few shots, we now propose to study our attempt at improving LoRA through a novel parametrization.

# 4 ALLoRA: ESCAPING LoRA'S FLAWS FOR FINE-TUNING

## 4.1 ADAPTIVE LEARNING

Section 3 summarized three flaws of LoRA, which we'll show can be addressed by a single solution. First we establish the underlying links among dropout, scale factor, and learning rate. Consider the LoRA finetuning of a single layer in (3), $f(\boldsymbol{x}) = (\boldsymbol{W}^* + \eta \boldsymbol{B} \boldsymbol{A})\boldsymbol{x}$. Following Hayou et al. (2024b), WLOG, we can further simply the model by assuming $\boldsymbol{W}^* = 0$, which is equivalent to defining $\tilde{\boldsymbol{y}} = \boldsymbol{y} - \boldsymbol{W}^* \boldsymbol{x}$ and rewrite the loss function by $\tilde{\boldsymbol{y}}$. Also assume $\eta = 1$, we have $f(\boldsymbol{x}) = \boldsymbol{B} \boldsymbol{A} \boldsymbol{x}$. The goal is the minimize loss $\mathcal{L}$ whose gradient is $g = \frac{\partial \mathcal{L}}{\partial (\boldsymbol{B} \boldsymbol{A})}$. $f(\boldsymbol{x}) \in \mathbb{R}^{n_1}$ is a column vector. Expanding it per row gives

$$(f(\boldsymbol{x}))_i = (\boldsymbol{B} \boldsymbol{A})_{i,:} \boldsymbol{x}, \quad i \in [n_1] \tag{4}$$

The effect of dropping out $(f(\boldsymbol{x}))_i$ for a given $i$ is equivalent to applying a per-row scaling factor $\eta_i = 0$ to $(\boldsymbol{B} \boldsymbol{A})_{i,:}$. Note that this is true only for $(\boldsymbol{B} \boldsymbol{A})_{i,:}$, the effect on $(\boldsymbol{B} \boldsymbol{A})_{j,:}, j \neq i$ is slightly different. Since $\frac{d\eta \cdot f(x)}{dx} = \eta \frac{df(x)}{x}$, $\eta_i = 0$ is implicitly applied to the $i$-th row of the gradient $g_i = (\frac{\partial \mathcal{L}}{\partial (\boldsymbol{B} \boldsymbol{A})})_{i,:}$, which is again a scaling factor applied to the learning rate $l$.

The observation reveals that both scaling factor and dropout are adaptions on LoRA output $f(x)$, and both have effects on gradient. We are inspired to formalize a general framework that subsumes both, within which we can use a principled approach to systematically discover novel solutions.

**Definition 2.** (Adaptive Learning) Consider a single layer linear model $f(\boldsymbol{x}) = \boldsymbol{B} \boldsymbol{A} \boldsymbol{x}$ with gradient $g(\boldsymbol{x}) = \frac{\partial \mathcal{L}(f(\boldsymbol{x}))}{\partial (\boldsymbol{B} \boldsymbol{A})}$. Let *Output Adaptor* be a function $f_o : \mathbb{R}^{n_1} \to \mathbb{R}^{n_1}$, *Gradient Adaptor* be a function $f_g : \mathbb{R}^{n_1 \times n_2} \to \mathbb{R}^{n_1 \times n_2}$. Define adapted output $\tilde{f}$ and adapted gradient $\tilde{g}$ by

$$\begin{cases} \tilde{f} = f_o \circ f \\ \tilde{g} = f_g \circ g \end{cases} \tag{5}$$

Adaptive learning is to use the adapted $\tilde{f}$ and $\tilde{g}$ in place of $f$ and $g$ respectively in the learning process.

Let $I : x \mapsto x$ be the identity function. Then a natural corollary is that all learning is adaptive learning (when $f_o = I$ and $f_g = I$). Note that $\mathcal{L}$ is a function of $f(\boldsymbol{x})$, hence $g(\boldsymbol{x}) = \frac{\partial (\mathcal{L} \circ f)}{\partial (\boldsymbol{B} \boldsymbol{A})}$. Use $\tilde{f}$ in place of $f$ defines a natural adapted gradient $\tilde{g} = \frac{\partial (\mathcal{L} \circ \tilde{f})}{\partial (\boldsymbol{B} \boldsymbol{A})} = \frac{\partial (\mathcal{L} \circ f_o \circ f)}{\partial (\boldsymbol{B} \boldsymbol{A})}$. When it's clear from the context, we omit $\tilde{g}$ if it's naturally defined by a non-trivial $\tilde{f}$.

## 4.2 ALLoRA

Under the Adaptive Learning framework, scaling factor is define by $f_o = \kappa : x \mapsto \eta x$.

$$\begin{cases} \tilde{f} = \kappa \circ f = \eta f \\ \tilde{g} = \kappa \circ f = \eta g \end{cases} \tag{6}$$

One idea to reduce the ripple effect while keeping the positive effect of scaling factor is to force $f_o = I$, while keeping $\tilde{g}$ intact, which is to use a larger learning rate $\eta \cdot l$. Nonetheless, a fixed learning rate cannot simultaneously achieve both fast escape from 0 and, once away from 0, measured discovery of optimal direction. We think a function that is inversely proportional to $||(\boldsymbol{BA})_{i,:}||$ is a good candidate to realize our idea. We use the function $1/\sqrt{||(\boldsymbol{BA})_{i,:}|| + 1/\eta^2}$ which reaches maximum at $||(\boldsymbol{BA})_{i,:}|| = 0$ and then tapers down when $||(\boldsymbol{BA})_{i,:}||$ increases (Figure 5).

Formally, ALLoRA is defined by

$$\begin{cases} \tilde{f} = I \circ f \\ \tilde{g}_i = 1/\sqrt{||(\boldsymbol{BA})_{i,:}|| + 1/\eta^2} \cdot g_i, \quad i \in [n_1] \end{cases}$$

where $\eta$ is a hyperparameter. Note that this does not introduce a new hyperparameter. We split learning rate into a constant base learning rate $l_b$ and $\eta$, and the effective learning rate is $\eta \cdot l_b$.

One more implementation detail is the backward pass computes, in addition to the gradients of $\boldsymbol{A}$ and $\boldsymbol{B}$, also the gradient of the input from layer below, and propagate which back to the layer below. We only modify the gradients of $\boldsymbol{A}$ and $\boldsymbol{B}$, but not that of the input. This helps further restrict the changes within each layer and reduce ripple effect.

To quickly verify our idea, we add probing code to trace the $L^2$ norm of row vectors of $\boldsymbol{BA}$. As shown in Figure 2 Right, adaptive learning rate escapes from 0 rapidly, the speed matches with LoRA with a learning rate that is $\eta \cdot l_b$. Then it finds an approriate level and enters measured discovery of optimal directions. LoRA with a learning rate lower than $\eta \cdot l_b$ can reach the same level, but at a much slower pace. The experiment is with **Snowflake Arctic XS** and **Rotten Tomatoes**.

## 4.3 A FAMILY OF ADAPTIVE SOLUTIONS

We adopt a principled approach to explore other reasonable designs that fall into the adaptive learning framework defined by Definition 2.

First notice that in $(f(\boldsymbol{x}))_i = (\boldsymbol{BA})_{i,:}\boldsymbol{x}$, $(f(\boldsymbol{x}))_i$ and $(\boldsymbol{BA})_{i,:}$ define each other. So instead of adapt the learning by $(\boldsymbol{BA})_{i,:}$, we can also adapt it by $(f(\boldsymbol{x}))_i$, which is *Output-Dependent*, or ALLoRA-OD, defined by

$$\begin{cases} \tilde{f} = I \circ f \\ \tilde{g}_i = 1/\sqrt{|(f(\boldsymbol{x}))_i| + 1/\eta^2} \cdot g_i, \quad i \in [n_1] \end{cases}$$

Note that $(f(\boldsymbol{x}))_i$ is a scalar, hence we use its absolute value. Qualitatively, ALLoRA-OD subjects to the stochastic noise in $\boldsymbol{x}$ because $(f(\boldsymbol{x}))_i = (\boldsymbol{BA})_{i,:}\boldsymbol{x}$. According to Smith et al. (2021), this type of stochastic noise is an implicit regularization. Our conjecture is that it may drag down the accuracy just as dropout does, and therefore ALLoRA-OD may not be as good as ALLoRA.

Given the link between learning rate and scaling factor, we may achieve similar effect by switching from adaptive learning rate to *Adaptive Scaling Factor*, or ASF-LoRA, defined by

$$\tilde{f}_i = 1/\sqrt{|(f(\boldsymbol{x}))_i| + 1/\eta^2} \cdot f_i, \quad i \in [n_1]$$

Note that $\tilde{g}$ is naturally defined by using $\tilde{f}$ in place of $f$. The potential downside is that it introduces ripple effect across layers, which may blur accuracy. And our conjecture is again ASF-LoRA may not be as good as ALLoRA.

One more caveat of ASF-LoRA is that we cannot merge $\boldsymbol{BA}$ back to $\boldsymbol{W}^*$ as we need to apply $f_o$ to the LoRA output.

**ALLoRA − LoRA**

| model:ds \ $\eta^2$ | 1.0 | 2.0 | 4.0 | Average |
|---|---|---|---|---|
| Qwen2-0.5b:bib | 0.48% | 0.06% | 0.12% | 0.22% |
| Qwen2-0.5b:emo | 1.59% | 0.76% | 0.32% | 0.89% |
| Qwen2-0.5b:rott | 0.45% | 0.36% | 0.94% | 0.58% |
| Artic-l:emo | -0.13% | -0.07% | 0.13% | -0.02% |
| Artic-l:rott | 0.58% | 0.09% | 0.21% | 0.29% |
| Artic-l:yelp | -0.07% | -0.02% | 0.14% | 0.02% |
| Open450m:emo | 1.09% | 0.00% | -0.06% | 0.34% |
| Open450m:rott | 0.24% | 0.21% | 0.06% | 0.17% |
| Open450m:yelp | -0.10% | 0.02% | 0.05% | -0.01% |
| Average | 0.46% | 0.16% | 0.21% | 0.28% |

**ALLoRA-0 − ALLoRA**

| model:ds \ $\eta^2$ | 1.0 | 2.0 | 4.0 | Average |
|---|---|---|---|---|
| Qwen2-0.5b:bib | 0.24% | 0.22% | -0.19% | 0.09% |
| Qwen2-0.5b:emo | 0.17% | -0.43% | 0.47% | 0.07% |
| Qwen2-0.5b:rott | 0.69% | 0.69% | -0.15% | 0.41% |
| Artic-l:emo | 0.05% | 0.29% | -0.17% | 0.06% |
| Artic-l:rott | -0.10% | -0.28% | 0.00% | -0.13% |
| Artic-l:yelp | -0.03% | 0.09% | 0.00% | 0.02% |
| Open450m:emo | -0.03% | -0.08% | 0.16% | 0.02% |
| Open450m:rott | -0.09% | 0.09% | -0.13% | -0.04% |
| Open450m:yelp | 0.04% | 0.05% | -0.01% | 0.03% |
| Average | 0.10% | 0.07% | 0.00% | 0.06% |

Figure 3: Accuracy gap between ALLoRA and LoRA. Each cell is an average of 5 runs. **Left**: AL-LoRA admits better accuracy than that of LoRA. **Right**: ALLoRA-0, the version without dropout, admits comparable accuracy than that of ALLoRA.

## 4.4 EMPIRICAL VALIDATION: PERCEPTION TASKS

Our first set of experiments gauges the performance of ALLoRA on perception tasks. Mainstream LLMs nowadays are mostly pretrained by next token prediction, which is good for generative tasks, but may not be a good fit for perception tasks such as Natural Language Understanding (NLU) and Sentiment Analysis (SA). In fact, we observe subpar accuracy when finetuning popular open-weight models for NLU and SA tasks (see Table 3). We hope to show that ALLoRA may help boost the accuracy for perception tasks.

For our experiments, we pick three midsized LLMs: **Qwen2-0.5B**, **Snowflake-Artic-L**, and **OpenELM-450M**, and four NLU and SA datasets: **Bias in Bios**, **Emotion**, **Rotten Tomatoes**, and **Yelp Review**. To demonstrate the stability of ALLoRA, we run the experiments with various $\eta^2 \in \{1, 2, 4\}$ with a fixed baseline learning rate $l_b = 1e - 4$. To be fair for LoRA, we run LoRA at learning rate $l \in \{1e-4, \sqrt{2}e-4, 2e-4\}$, respectively. We finetune for 2 epochs. Each experiment is run 5 times and we report average final accuracy. Figure 3 Left shows the accuracy gap between ALLoRA and LoRA, where positive numbers indicate ALLoRA has better accuracy. The result shows that ALLoRA in general admits better accuracy over plain LoRA. Average improvement over all cases is $0.3\%$.

In the experiment, we use a dropout rate 0.05 for both ALLoRA and LoRA. We also run ALLoRA-0, the version of ALLoRA with 0 dropout rate. Figure 3 Right shows that there is no evident difference between ALLoRA and ALLoRA-0, matching our theoretical result from 3.1.

## 4.5 EMPIRICAL VALIDATION: COMMONSENSE REASONING

We also compare ALLoRA with DoRA (Liu et al. (2024a)), a recent LoRA variant that demonstrated superb performance over a range of PEFT methods. Since DoRA results are universally better than other PEFT methods, we only compare ALLoRA to DoRA. We run experiments on **LLaMA-7B**, **LLaMA2-7B**, and **LLaMA3-8B** on 8 **Commonsense** tasks. Following DoRA's setup, for each model, we run both ALLoRA and ALLoRA-0 with LoRA rank $r \in \{16, 32\}$ and for 3 epochs. Table 1 shows that for all cases, either ALLoRA or ALLoRA-0 has the best average accuracy. On average, ALLoRA and ALLoRA-0 each boosted accuracy by $0.3\%$ over DoRA.

Note that we run experiments with various $\eta^2 \in \{1, 2, 4\}$ and report the best accuracy[1], this follows DoRA's practices to run with various learning rate $l \in \{1e - 4, 2e - 4\}$ and report the best.

In Table 1 we also report the number of trainable parameters as a percentage of the number of pretrained parameters. Since ALLoRA does not introduce additional trainable parameters, its trainable parameters are slightly lower than that of DoRA.

---

[1]Weights of ALLoRA and ALLoRA-0 will be publicly shared in near future.

Table 1: Accuracy comparison of LLaMA 7B, LLaMA2 7B, and LLaMA3 8B between ALLoRA and DoRA on eight commonsense reasoning datasets. DoRA results are taken from Liu et al. (2024a). ALLoRA-0 is the version of ALLoRA with 0 dropout rate.

| Model LoRA Rank | Method | # Params % | BoolQ | PIQA | SIQA | HSwag | WGrande | ARC-e | ARC-c | OBQA | Avg. |
|---|---|---|---|---|---|---|---|---|---|---|---|
| LLaMA-7B 16 | DoRA | 0.43 | 70.0 | 82.6 | 79.7 | 83.2 | 80.6 | 80.6 | 65.4 | 77.6 | 77.5 |
| | ALLoRA (ours) | 0.41 | 69.4 | 82.7 | 78.3 | 84.8 | 80.0 | 80.9 | 65.7 | 79.2 | **77.6** |
| | ALLoRA-0 (ours) | 0.41 | 69.2 | 80.8 | 78.5 | 83.9 | 81.1 | 80.8 | 65.2 | 78.2 | 77.2 |
| LLaMA-7B 32 | DoRA | 0.84 | 69.7 | 83.4 | 78.6 | 87.2 | 81.0 | 81.9 | 66.2 | 79.2 | 78.4 |
| | ALLoRA (ours) | 0.83 | 70.0 | 82.3 | 78.1 | 84.6 | 82.2 | 81.0 | 67.9 | 81.0 | **78.4** |
| | ALLoRA-0 (ours) | 0.83 | 70.2 | 82.6 | 78.6 | 83.8 | 81.1 | 81.0 | 66.3 | 82.6 | 78.3 |
| LLaMA2-7B 16 | DoRA | 0.43 | 72.0 | 83.1 | 79.9 | 89.1 | 83.0 | 84.5 | 71.0 | 81.2 | 80.5 |
| | ALLoRA (ours) | 0.41 | 71.7 | 83.7 | 79.5 | 91.4 | 82.4 | 84.3 | 69.2 | 81.2 | 80.4 |
| | ALLoRA-0 (ours) | 0.41 | 72.4 | 83.9 | 80.0 | 90.8 | 83.0 | 84.7 | 71.3 | 80.2 | **80.8** |
| LLaMA2-7B 32 | DoRA | 0.84 | 71.8 | 83.7 | 76.0 | 89.1 | 82.6 | 83.7 | 68.2 | 82.4 | 79.7 |
| | ALLoRA (ours) | 0.83 | 72.2 | 83.1 | 79.6 | 91.2 | 84.5 | 84.5 | 71.0 | 80.0 | 80.8 |
| | ALLoRA-0 (ours) | 0.83 | 72.3 | 83.8 | 79.3 | 91.4 | 83.0 | 85.0 | 71.2 | 82.2 | **81.0** |
| LLaMA3-8B 16 | DoRA | 0.35 | 74.5 | 88.8 | 80.3 | 95.5 | 84.7 | 90.1 | 79.1 | 87.2 | 85.0 |
| | ALLoRA (ours) | 0.35 | 75.2 | 88.9 | 80.8 | 95.6 | 84.7 | 90.2 | 80.6 | 85.8 | 85.2 |
| | ALLoRA-0 (ours) | 0.35 | 74.5 | 89.1 | 80.4 | 95.5 | 85.8 | 90.7 | 80.3 | 86.0 | **85.3** |
| LLaMA3-8B 32 | DoRA | 0.71 | 74.6 | 89.3 | 79.9 | 95.5 | 85.6 | 90.5 | 80.4 | 85.8 | 85.2 |
| | ALLoRA (ours) | 0.70 | 74.5 | 88.9 | 81.8 | 95.9 | 86.3 | 90.4 | 80.5 | 87.6 | **85.8** |
| | ALLoRA-0 (ours) | 0.70 | 75.1 | 88.7 | 81.8 | 95.8 | 85.4 | 91.0 | 81.1 | 86.6 | 85.7 |

## 5 ABLATION STUDY

Using the same setup in 4.4, we run experiments with ALLoRA-OD, the output-dependent variant, and ASF-LoRA, the adaptive scaling factor variant. We also run LoRA with comparable fixed scaling factors to form an objective baseline for ASF-LoRA.

### 5.1 ALLoRA-OD

Figure 4 Top Left shows the accuracy gap between ALLoRA and ALLoRA-OD. A positive number indicates that ALLoRA has better accuracy. Overall speaking, ALLoRA has better accuracy than ALLoRA-OD. But the difference is moderate, as average improvement over all cases is $0.3\%$.

The result matches our conjecture that stochastic noise experienced by ALLoRA-OD might have dragged down accuracy at early epochs.

### 5.2 ASF-LoRA AND LoRA WITH FIXED SCALING FACTOR

Since a scaling factor on output is implicitly also a scaling factor on gradient, we use the same $\eta$ when comparing between ALLoRA and ASF-LoRA, i.e., the gradient adaptor $f_g$ in ALLoRA and the output adaptor $f_o$ use the same $\eta$.

Figure 4 Top Right shows the accuracy gap between ALLoRA and ASF-LoRA. A positive number indicates that ALLoRA has better accuracy. ALLoRA has a significant advantage over ASF-LoRA as average improvement over all cases is $1.0\%$. Since we know that ALLoRA-OD is only slightly worse than ALLoRA, the evidence leans toward that Adaptive Learning Rate is in general a better solution family than Adaptive Scaling Factor.

We also run LoRA at comparable fixed scaling factors $\frac{\alpha}{r} \in \{1, \sqrt{2}, 2\}$. The results, as shown in Figure 4 Bottom, show that

- ASF-LoRA is not a competitive method, as over half of cases see ASF-LoRA's accuracy significantly lower than LoRA with a comparable fixed scaling factor (positive numbers in Figure 4 Bottom Right).

**ALLoRA − ALLoRA-OD**

| model:ds \ $\eta^2$ | 1.0 | 2.0 | 4.0 | Average |
|---|---|---|---|---|
| Qwen2-0.5b:emo | 0.27% | 0.37% | 0.33% | 0.32% |
| Qwen2-0.5b:rott | 1.33% | 1.13% | 0.88% | 1.11% |
| Artic-l:emo | -0.31% | 0.01% | 0.11% | -0.06% |
| Artic-l:rott | 0.83% | 0.28% | 0.04% | 0.38% |
| Open450m:emo | -0.24% | 0.15% | 0.00% | -0.03% |
| Open450m:rott | 0.09% | -0.08% | 0.23% | 0.08% |
| Average | 0.33% | 0.31% | 0.26% | 0.30% |

**ALLoRA − ASF-LoRA**

| model:ds \ $\eta^2$ | 1.0 | 2.0 | 4.0 | Average |
|---|---|---|---|---|
| Qwen2-0.5b:emo | 0.78% | 1.32% | 0.55% | 0.88% |
| Qwen2-0.5b:rott | 0.21% | 0.51% | 1.43% | 0.71% |
| Artic-l:emo | -0.37% | 0.49% | 1.11% | 0.41% |
| Artic-l:rott | 0.26% | 1.26% | 1.01% | 0.84% |
| Open450m:emo | 2.63% | 3.75% | 1.63% | 2.67% |
| Open450m:rott | 0.62% | 0.73% | 0.81% | 0.72% |
| Average | 0.69% | 1.34% | 1.09% | 1.04% |

**ALLoRA − LoRA $\times\eta$**

| model:ds \ $\eta^2$ | 1.0 | 2.0 | 4.0 | Average |
|---|---|---|---|---|
| Qwen2-0.5b:emo | 1.59% | 1.91% | 1.37% | 1.62% |
| Qwen2-0.5b:rott | 0.45% | 0.21% | 2.14% | 0.93% |
| Artic-l:emo | -0.13% | 0.61% | 0.59% | 0.36% |
| Artic-l:rott | 0.58% | 0.71% | 0.94% | 0.74% |
| Open450m:emo | 1.09% | 1.24% | 1.02% | 1.12% |
| Open450m:rott | 0.24% | 0.56% | 0.53% | 0.44% |
| Average | 0.64% | 0.87% | 1.10% | 0.87% |

**LoRA $\times\eta$ − ASF-LoRA**

| model:ds \ $\eta^2$ | 1.0 | 2.0 | 4.0 | Average |
|---|---|---|---|---|
| Qwen2-0.5b:emo | -0.81% | -0.59% | -0.82% | -0.74% |
| Qwen2-0.5b:rott | -0.24% | 0.30% | -0.71% | -0.22% |
| Artic-l:emo | -0.24% | -0.12% | 0.52% | 0.05% |
| Artic-l:rott | -0.32% | 0.54% | 0.08% | 0.10% |
| Open450m:emo | 1.54% | 2.51% | 0.61% | 1.55% |
| Open450m:rott | 0.38% | 0.17% | 0.28% | 0.28% |
| Average | 0.05% | 0.47% | -0.01% | 0.17% |

Figure 4: Accuracy gap between ALLoRA and other adaptive approaches. Each cell is an average of 5 runs. **Top left**: ALLoRA admits better accuracy than that of ALLoRA-OD, where learning rate is LoRA **O**utput-**D**ependent. **Top right**: ALLoRA admits better accuracy than that of ASF-LoRA, where an **A**daptive **S**cale **F**actor is applied to LoRA output. **Bottom left**: ALLoRA admits better accuracy than that of LoRA with a fixed scaling factor. **Bottom right**: Adaptive scale factor is not better than a fixed scale factor.

- ALLoRA is significantly better than LoRA with a comparable fixed scaling factor, as average improvement over all cases is $0.9\%$.

# 6 CONCLUSION AND FUTURE WORK

This paper identifies three major flaws of LoRA, namely dropout, poor optimization landsacpe, and scaling factor. We conducted principled analysis and proved that dropout is not a must-have in the finetuning regime. After uncovering the hidden connection between dropout, scaling factor, and learning rate, we proposed a unified adaptive learning framework to address them all. Motivated by which, we proposed a novel LoRA variant: ALLoRA. Empirical results show that ALLoRA admits better accuracy than plain LoRA over multiple backbones, datasets, and learning rates; and better accuracy than recent successful LoRA variants such as DoRA. Ablation study shows that ALLoRA is the optimal in a family of adaptive methods.

Below we list a few interesting research directions and invite researchers to explore the frontier opened-up by our research:

- The adaptive learning framework introduced by this paper is generic and may find broad applications beyond LoRA. Other use cases such as pretraining may not have the constraints that the weight matrix must be initialized at $0$. But they may have other types of constraints, which may be solved by adaptive learning with a different adaptor function.

- Within the LoRA use case, we only examine one particular adaptor function, there could be other adaptor functions that have superior performance.

- We only provide empirical evidence that ALLoRA admits better accuracy. Theoretical guarantee is needed, especially for the convoluted case where the base model has multiple layers.

- Starting from $0$ weights may avoid the lottery ticket hypothesis (Frankle & Carbin (2019)), for good or bad, where adaptive learning rate can be a handy tool.

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

Table 2: Reprise of Figure 2 Left depicting that the epoch at which the LoRA fine-tuned model reaches the best accuracy increases with the Dropout rate, i.e., the larger the probability to drop dimensions, the more regularization is applied and the better the final performance–but only after very long fine-tuning.

| dropout | 0.0 | 0.05 | 0.1 | 0.2 | 0.4 |
|---|---|---|---|---|---|
| Acc. at epoch=3 | 77.92 | 77.96 | 77.68 | 77.44 | 77.22 |
| Acc. at epoch=10 | 78.00 | 79.95 | 80.50 | 80.77 | 80.65 |
| Max acc. | 79.81 | 80.27 | 80.54 | 80.78 | 80.65 |
| Epoch of max acc. | 6 | 8 | 8 | 9 | 10 |

## A  PROOF OF UPPER BOUND

*Proof.*

$$
\mathbb{E}\left[\left|\frac{1}{N}\sum_{n=1}^{N}\|\boldsymbol{Y}-(\boldsymbol{XW})\odot\boldsymbol{V}_n\|_F^2 - \mathbb{E}\left[\|\boldsymbol{Y}-(\boldsymbol{XW})\odot\boldsymbol{V}\|_F^2\right]\right|\right]
$$

$$
=\mathbb{E}\left[\sqrt{\left(\frac{1}{N}\sum_{n=1}^{N}\|\boldsymbol{Y}-(\boldsymbol{XW})\odot\boldsymbol{V}_n\|_F^2 - \mathbb{E}\left[\|\boldsymbol{Y}-(\boldsymbol{XW})\odot\boldsymbol{V}\|_F^2\right]\right)^2}\right]
$$

$$
\leq\sqrt{\mathbb{E}\left[\left(\frac{1}{N}\sum_{n=1}^{N}\|\boldsymbol{Y}-(\boldsymbol{XW})\odot\boldsymbol{V}_n\|_F^2 - \mathbb{E}\left[\|\boldsymbol{Y}-(\boldsymbol{XW})\odot\boldsymbol{V}\|_F^2\right]\right)^2\right]}
$$

$$
=\sqrt{\mathbb{E}\left[\left(\frac{1}{N}\sum_{n=1}^{N}\|\boldsymbol{Y}-(\boldsymbol{XW})\odot\boldsymbol{V}_n\|_F^2\right)^2\right] - \mathbb{E}\left[\|\boldsymbol{Y}-(\boldsymbol{XW})\odot\boldsymbol{V}\|_F^2\right]^2}
$$

$$
=\sqrt{\mathbb{E}\left[\left(\frac{1}{N}\sum_{n=1}^{N}\|\boldsymbol{Y}-(\boldsymbol{XW})\odot\boldsymbol{V}_n\|_F^2\right)^2\right] - \mathbb{E}\left[\|\boldsymbol{Y}-(\boldsymbol{XW})\odot\boldsymbol{V}\|_F^2\right]^2}
$$

$$
=\sqrt{\frac{1}{N^2}\sum_{n=1}^{N}\mathbb{E}\left[\|\boldsymbol{Y}-(\boldsymbol{XW})\odot\boldsymbol{V}_n\|_F^4\right] - \mathbb{E}\left[\|\boldsymbol{Y}-(\boldsymbol{XW})\odot\boldsymbol{V}\|_F^4\right]}
$$

$$
=\sqrt{\frac{1}{N}Var\left[\|\boldsymbol{Y}-(\boldsymbol{XW})\odot\boldsymbol{V}\|_F^2\right]}
$$

$$
=\frac{Std\left[\|\boldsymbol{Y}-(\boldsymbol{XW})\odot\boldsymbol{V}\|_F^2\right]}{\sqrt{N}}
$$

□

## B  FINETUNING ACCURACY AT VARIOUS DROPOUT RATES

Table 2 contains accuracy of various dropout rates at different number of epochs.

## C  ADAPTIVE FUNCTION

Figure 5 is an adaptive function that provides output value when $|x| = 0$, and then tapers down when $|x| > 0$.

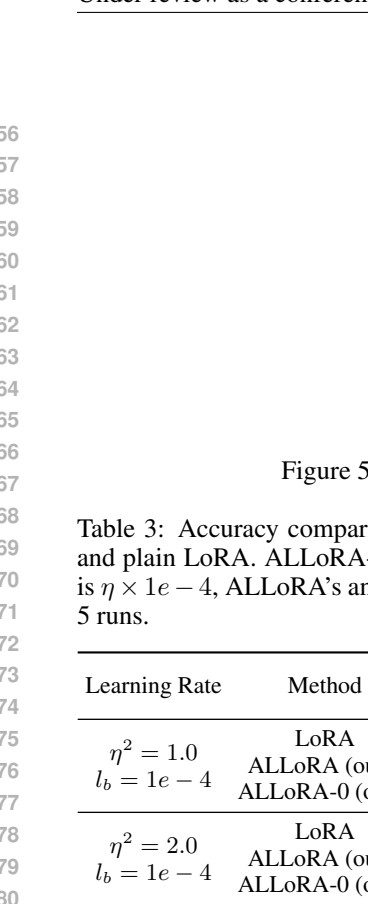

Figure 5: Adaptive function $1/\sqrt{|x| + 1/\eta^2}$ for $\eta^2 = 1, 2, 4$.

Table 3: Accuracy comparison of various models, datasets, and learning rates between ALLoRA and plain LoRA. ALLoRA-0 is the version of ALLoRA with $0$ dropout rate. LoRA's learning rate is $\eta \times 1e - 4$, ALLoRA's and ALLoRA-0's base learning rate is $1e - 4$. Each cell is an average over 5 runs.

| Learning Rate | Method | Qwen2-0.5B | | | Snowflake-Artic-L | | | OpenELM-450M | | |
|---|---|---|---|---|---|---|---|---|---|---|
| | | b-in-b | emotion | rotten | emotion | rotten | yelp | emotion | rotten | yelp |
| $\eta^2 = 1.0$ $l_b = 1e - 4$ | LoRA | 70.81 | 35.59 | 53.19 | 87.06 | 77.71 | 63.61 | 84.98 | 87.95 | 71.41 |
| | ALLoRA (ours) | 71.29 | 37.18 | 53.64 | 86.93 | **78.29** | 63.55 | **86.07** | **88.20** | 71.32 |
| | ALLoRA-0 (ours) | **71.53** | **37.35** | **54.33** | **86.98** | 78.19 | 63.52 | 86.04 | 88.11 | 71.36 |
| $\eta^2 = 2.0$ $l_b = 1e - 4$ | LoRA | 73.70 | 37.76 | 54.32 | 88.30 | 79.34 | 64.26 | 90.01 | 88.74 | 71.62 |
| | ALLoRA (ours) | 73.77 | **38.52** | 54.67 | 88.23 | **79.44** | 64.24 | **90.01** | 88.95 | 71.63 |
| | ALLoRA-0 (ours) | **73.99** | 38.09 | **55.37** | **88.52** | 79.16 | **64.33** | 89.93 | **89.04** | **71.68** |
| $\eta^2 = 4.0$ $l_b = 1e - 4$ | LoRA | 75.60 | 38.68 | 54.90 | 88.95 | 80.04 | 64.76 | 91.33 | 89.55 | 71.79 |
| | ALLoRA (ours) | **75.72** | 39.00 | **55.83** | **89.08** | 80.24 | 64.61 | 91.27 | **89.61** | **71.83** |
| | ALLoRA-0 (ours) | 75.52 | **39.47** | 55.68 | 88.91 | **80.24** | **64.76** | **91.43** | 89.47 | 71.82 |

## D  PERCEPTION TASKS

Table 3 shows the accuracy data of all of our experiments on perception tasks. Each cell is an average of 5 runs. ALLoRA is universally better than LoRA in terms of accuracy.

## E  ABLATION

Table 4 shows the accuracy data of all of our ablation study on perception tasks. Each cell is an average of 5 runs. ALLoRA is universally better than the rest in the family.

Table 4: Accuracy comparison of various models, datasets, and learning rates between ALLoRA and other adaptive approaches. for adaptive learning rate approaches, i.e., ALLoRA and ALLoRA-OD, base learning rate is $1e-4$. For adaptive scaling factor, i.e., ASF-LoRA, learning rate is fixed at $1e-4$, an adaptive scaling factor $1/\sqrt{|x|+1/\eta^2}$ is applied. For LoRA, a fixed scaling factor $\eta$ is applied, and learning rate is fixed at $1e-4$. Each cell is an average over 5 runs.

| $\eta^2$ | Method | Qwen2-0.5B | | Snowflake-Artic-L | | OpenELM-450M | |
| | | emotion | rotten | emotion | rotten | emotion | rotten |
|---|---|---|---|---|---|---|---|
| 1.0 | ALLoRA (ours) | **37.18** | **53.64** | 86.93 | **78.29** | 86.07 | **88.20** |
| | ALLoRA-OD | 36.91 | 52.31 | 87.24 | 77.47 | 86.31 | 88.11 |
| | ASF-LoRA | 36.40 | 53.43 | 87.30 | 78.03 | 83.44 | 87.58 |
| | LoRA $\times\eta$ | 35.59 | 53.19 | 87.06 | 77.71 | 84.98 | 87.95 |
| 2.0 | ALLoRA (ours) | **38.52** | **54.67** | **88.23** | **79.44** | **90.01** | 88.95 |
| | ALLoRA-OD | 38.15 | 53.55 | 88.22 | 79.16 | 89.86 | 89.02 |
| | ASF-LoRA | 37.20 | 54.17 | 87.74 | 78.18 | 86.26 | 88.22 |
| | LoRA $\times\eta$ | 36.61 | 54.47 | 87.62 | 78.72 | 88.77 | 88.39 |
| 4.0 | ALLoRA (ours) | **39.00** | **55.83** | **89.08** | **80.24** | **91.27** | **89.61** |
| | ALLoRA-OD | 38.67 | 54.95 | 88.97 | 80.21 | 91.27 | 89.38 |
| | ASF-LoRA | 38.45 | 54.41 | 87.97 | 79.23 | 89.64 | 88.80 |
| | LoRA $\times\eta$ | 37.63 | 53.70 | 88.49 | 79.31 | 90.25 | 89.08 |

# F  CODE

```python
class ALLoRA(torch.autograd.Function):
  rsq_scale = 1. / 4.  # 1 / \eta^2

  @staticmethod
  def forward(ctx, input_x, weight_A, weight_B):
    output = input_x @ weight_A.t() @ weight_B.t()
    norms = torch.norm(weight_B @ weight_A, dim=1)
    ctx.save_for_backward(input_x, weight_A, weight_B, norms)
    return output

  @staticmethod
  def backward(ctx, grad_output):
    input_x, weight_A, weight_B, norms = ctx.saved_tensors
    accelerate = 1. / torch.sqrt(norms + LinearLayer2.rsq_scale)
    grad_input = grad_output @ weight_B @ weight_A
    temp = grad_output.mul(accelerate) @ weight_B
    temp = torch.transpose(temp, 1, 2)
    grad_weight_A = temp @ input_x
    temp = grad_output.mul(accelerate).transpose(1, 2)
    grad_weight_B = temp @ (input_x @ weight_A.t())
    return grad_input, grad_weight_A, grad_weight_B
```

Listing 1: ALLoRA Code

