# OpenReview forum: "ALLoRA: Adaptive Learning Rate Mitigates LoRA Fatal Flaws"
_ICLR.cc/2025/Conference — Submitted to ICLR 2025_

### Official Review · Reviewer_oeVZ · 2024-10-29

**Soundness:** 2
**Presentation:** 2
**Contribution:** 2
**Rating:** 3
**Confidence:** 4

**Summary:**

Motivated by the identification of three core limitations in the standard Low-Rank Adaptation (LoRA) method, this paper introduces ALLoRA, a variant designed to address and overcome these flaws. The authors first conduct theoretical justifications, utilizing toy examples to elucidate the issues inherent in LoRA, particularly within the context of fine-tuning with limited training steps. Building upon these analyses, the paper proposes a solution in the form of LoRA with Adaptive Learning Rate (ALLoRA). Empirical experiments are then conducted to validate the effectiveness of ALLoRA, demonstrating its performance compared to both the standard LoRA method and other recent variants.

**Strengths:**

1. This paper is generally well-motivated, where the identified three flaws of standard LoRA may be of independent interest for readers.
2. This paper provide thorough theoretical explanations, trying to support the three core limitations of standard LoRA. These theoretical justifications may also be of independent interest for readers.

**Weaknesses:**

1. In general, the theoretical justifications in this paper do not fully support the three flaws of standard LoRA as mentioned. Consequently, the motivation of this paper may not be convincing, at least to me.
    - For the first flaw, the connection between expectation and training steps may be incorrect. The expectation is over the randomness of V, whereas infinite training steps imply that W approaches the closed-form solution W*. To demonstrate that dropout is not suitable for fine-tuning with limited training steps, the authors need to prove that infinite training steps can lead to the expectation of the results. More importantly, how is "limited training steps" defined? Dropout with large rates outperforms that with small rates even for 8-epoch training, which is also considered limited compared to pre-training.
    - For the second flaw, it is unclear how the degenerate optimization landscape is related to the zero-initialization of LoRA. More importantly, why does the landscape become degenerate as the training set size (n) decreases? As far as I can see, this is related to X^TX, which may remain the same even after n decreases.
    - For the third flaw, it is unclear how the output's potential exponential growth with respect to the number of layers is connected to the performance of LoRA. If this connection is not made clear, we cannot conclude that it is harmful for the final fine-tuning performance.

2. Supposing the "flaws" are true, the paper fails to explain why ALLoRA is a good or necessary way to overcome them. More specifically, one could also directly remove dropout, the scaling factor, and so on to address these issues.

3. This paper primarily compares with DoRA and lacks comparisons with other more recent LoRA techniques, such as those in (Hayou et al., 2024a, 2024b), rsLoRA, Flexora, and others. Importantly, the improvement is quite marginal.

4. This paper lacks an ablation study to show how ALLoRA can help overcome each flaw independently. This is crucial for understanding the efficacy of ALLoRA.

5. The citation format needs to be refined. There are many places where the authors should use \citep to cite a paper, but they have applied \citet to cite the authors.

**Questions:**

- For lines 193-197, the authors need to provide more elaboration on the derivation. Specifically, it is unclear how ( p ) must be equal to ( 1/(1+\lambda) ). Additionally, there is no explanation to ensure that the final solution of ( W ) should be equivalent for the cases of dropout and Tikhonov regularization.
- What is the difference between ALLoRA and adaptive gradient methods?

---

### Official Review · Reviewer_RUEL · 2024-10-31

**Soundness:** 2
**Presentation:** 2
**Contribution:** 1
**Rating:** 3
**Confidence:** 4

**Summary:**

This work presents a novel low-rank adaptation algorithm that tries to resolve the issues of dropout, poor optimization landscape, and scaling factor. Specifically, the authors propose a dropout-free, scaling-free LoRA with adaptive learning rate, termed ALLoRA. By scaling the per sample and per parameter gradients with a coefficient inversely proportional to parameters' $l_2$ norm, ALLoRA alleviates the above issues. Also, ALLoRA can remove two hyperparameters from LoRA, the scaling factor and the dropout rate. To validate the proposed approach, the authors utilize different models and datasets to show the competitive performance of ALLoRA with the comparison with LoRA and DoRA.

**Strengths:**

1. The investigated topic in this paper is quite interesting and critical as fine-tuning large models have now been very popular in many domain applications.
2. The proposed method is technically precise and straightforward, which may ease the real deployments.
3. This work is well motivated and the technical issues are well studied.
4. The proposed method is validated through extensive results.

**Weaknesses:**

1. The novelties in this work are quite marginal. Although the authors have proposed the novel LoRA variant, it lacks of in-depth theoretical analysis on ALLoRA. The authors should try to have more comprehensive technical analysis why ALLoRA allows for improvement compared to vanilla LoRA and CoRA.

2. In Figure 1, the authors use two datasets MNIST and CIFAR10 to demonstrate the flaw of dropout. They plot the distribution of standard deviation of gradients. However, during the training, they consider a single mini-batch, instead of the regular one pass over the whole dataset in practice. This may cause the worse variance of the gradient during the training. That way, the claim that this is attributed to Dropout does not make much sense in this context. Also, in the caption, they have the conclusion "while the average variance of the gradient decreases slightly during training, the worst case increases, hence leading to unstable training in finetuning regimes". But this conclusion is not well supported by the two plots.

3. In Figure 2, I am confused how the authors use these two plot to demonstrate the dropout can hurt the fine-tuning of large models. Particularly, what is the purpose to show the right plot? Shouldn't it be a direct comparison between LoRA and ALLoRA in terms of accuracy, indicating that without dropout, ALLoRA can perform better? Also, the authors claim that 10 epochs is already a large number of finetuning iterations for practical scenarios. How to justify this? Is there any a standard to tell what number of iterations is large in finetuning?

4. In Section 3.2, the authors would like to show the second flaw of finetuning LLMs when the zero-initialization is used, which can cause poor optimization landscape. But how can we see this issue from this section? The authors claim that as the training set size decreases, the optimization landscape for $V$ will become degenerate. I am confused about this. Just from the Eqs. (1) and (2), how did the authors make this claim? They should at least show an example for the poor optimization landscape.

5. In Section 3.3, from Line 275 - 280, the derivation here is confusing. I understand that the authors would like have $(1+\eta)^L$ in the final term. However, when they define $C$ , how can the second inequality follow from? I believe $C$ is a consecutive product among all matrix norms of $W^*_1$,...,$W^*_L$. Also, can the authors tell in the paper why they need $\bar{m}$ in the paper? Please derive step by step clearly in the paper.

6. Notations are confusing. From the beginning, the authors didn't really define $W^*$. Then in line 306, they say $l$ is a learning rate, but in previous context, $l$ indicates the layer. Also, what does it mean by $f_o=\kappa:x\mapsto\eta x$? Please check thoroughly all notations in the paper to make sure they are clearly and properly defined.

7. In line 334, the authors mention that "We think a function that is inversely proportional to $||(BA)_{i,:}||$ is a good candidate to realize our idea". Why? You need justification here, not just saying that.

8. The Adaptive Learning in this work seems to have scaling factor in the functional, instead of directly applying to model parameters. However, to me, their effects are similar. That way, what is the exact difference and novelty in this study ALLoRA brings, compared to LoRA. Also, I didn't see how the second issue, poor optimization landscape has been addressed through ALLoRA.

9. The experimental results are not that promising. Though the authors present many results, from Section 4.4, the average improvement over all cases is 0.3%, which is even within the standard deviation, in my opinion. The same to Section 4.5. With such marginal performance improvement, what values does ALLoRA bring about?

10. Overall, though the topic in this work looks interesting, it requires a substantial amount of efforts from the authors to make it technically solid and sound, including more in-depth theoretical analysis, clarification on technical discussion, and more convincing experimental results.

**Questions:**

Please see the above comments for questions.

---

### Official Review · Reviewer_RkUw · 2024-11-04

**Soundness:** 2
**Presentation:** 3
**Contribution:** 2
**Rating:** 3
**Confidence:** 4

**Summary:**

The paper identifies three key limitations of Low-Rank Adaptation (LoRA) in the context of fine-tuning large language models: the ineffectiveness of dropout for short training epochs, poor optimization landscape due to zero initialization, and problematic interactions due to the scaling factor. The authors propose ALLoRA, which addresses these issues through an adaptive learning rate approach that scales gradients inversely proportional to L2 norm. While the paper presents interesting ideas, there are significant concerns about the depth of analysis and justification of claims.

**Strengths:**

- Identifies potential limitations in the current LoRA approach.
- Novel adaptive learning rate solution that addresses these issues.
- Reduction in hyperparameters while maintaining or improving performance.
- Clear ablation studies demonstrating the effectiveness of different components.
- Comprehensive empirical validation across different models and tasks.

**Weaknesses:**

- Their analysis relies on the fact that we only fine-tune for a small number of epochs. This is highly subjective as the number of epochs required for a good fine tuning depends on the model and the dataset.
- Limited theoretical analysis of the adaptive learning rate’s convergence properties.
- The ripple effect argument lacks mathematical rigor, only establishing upper bounds without lower bounds or empirical validation.
- Use of simplified models may not accurately represent LLM fine-tuning dynamics.
- Absence of standard deviations in results makes it difficult to assess statistical significance.
- Lack of comparison with well-established benchmarks (e.g., GLUE).
- No comparison with similar approaches like LoRA+.
- Arbitrary choice of epoch numbers without clear justification.
- No discussion of computational overhead compared to standard LoRA.
- Unclear visualization of dropout’s impact in initial epochs.
- Limited justification for why/how ALLoRA ensures faster or more reliable convergence.

**Questions:**

- How does the computational cost of ALLoRA compare to standard LoRA?
- Are there any scenarios where the adaptive learning rate approach might be disadvantageous?
- How sensitive is the method to the choice of η² hyperparameter?
- Could the approach be combined with other recent LoRA variants for additional benefits?
- Why was the GLUE benchmark not included in the evaluation?
- How does ALLoRA compare to LoRA+ in terms of performance?
- What criteria were used to determine the optimal number of epochs for different datasets? For instance, RTE requires 50 epochs for good performance (according to literature), would your framework still apply?
- Can you provide statistical significance analysis of the performance improvements in terms of standard deviations?

---

### Official Review · Reviewer_PH2e · 2024-11-04

**Soundness:** 2
**Presentation:** 1
**Contribution:** 2
**Rating:** 3
**Confidence:** 4

**Summary:**

The paper points out three limitations of existing LoRA methods, unnecessary Dropout for short fine-tuning session, suboptimal zero initialization that results in poor optimization landscape, and uniform scaling factor across layers. Authors resolve these problems by removing Dropout regularization and the scaling factor out of the LoRA design. Additionally, they add an adaptive learning rate strategy that adaptively scales the update inversely proportional to the weight norm. With these modifications, ALLoRA improves performance over LoRA and other recent variants.

**Strengths:**

The paper suggest simple yet efficient implementation of LoRA methods by strategically adjusting the existing elements like learning rates, dropout, and scaling factors. They also provide empirical validations on the benefits of their design choice over other LoRA variants such as DoRA.

**Weaknesses:**

1. ALLoRA relies on a heuristic approach that seems more like a refined hyper-parameter adjustments rather than a fundamental improvement in LoRA architecture. The core changes such as adapting learning rates based on parameter norms, removing Dropout, and eliminating the scaling factor can be viewed as incremental modifications to existing techniques, not a novel structural design.


2. Limited experiments in terms of datasets, baseline models, comparison with other recent LoRA variants.
- Their experimental setup doesn't align with the standard setting. No GLUE benchmark experiments and the downstream task is limited to the commonsense reasoning task on LLaMA variants.
- Missing comparison to the various SOTA LoRA variants, only DoRA is used to compare the performance of ALLoRA on commonsense reasoning datasets.


3. Ablation study done in non-ideal settings.
- Ablation study not done on the same settings used for performance report, only done on midsized LLMs Qwen2-0.5B, Snowflake-Artic-L, and OpenELM-450M.

**Questions:**

- I suggest expanding the experimental setup to include standard benchmarks like GLUE and additional SOTA LoRA variants beyond DoRA, for a broader evaluation of ALLoRA’s effectiveness.

- Seems like ALLoRA uses same dropout rate as LoRA, 0.05, and 0 dropout rate is suboptimal even under short fine-tuning paradigm, as shown in Table 2. Any explanation why?

- Could you conduct ablation studies on the same model and dataset configurations used in the primary performance evaluations to ensure consistency across experimental settings?

---

### Official Review · Reviewer_bDRt · 2024-11-04

**Soundness:** 2
**Presentation:** 3
**Contribution:** 3
**Rating:** 3
**Confidence:** 4

**Summary:**

The authors point out three core issues when fine-tuning LLM with LoRA: i) the instability of dropout in short-term training such as fine-tuning, ii) poor optimization landscape due to the zero initialization of the adapter, and iii) the scaling factor causes nonlinear interactions between LoRA layers. To solve these issues, the authors remove two hyperparameters, dropout and scaling factor, and introduce an adaptive learning rate. To address this issue, the authors removed two hyperparameters, dropout and scaling factor, and instead introduced an adaptive learning rate approach. This approach helps the model quickly move away from the initial zero state and gradually reduces the learning rate as training progresses, promoting stable convergence.

**Strengths:**

- The novel aspect lies in identifying and addressing the issues commonly accepted in LoRA training, such as dropout, initialization with zero, and the scaling factor.
- By removing two types of hyperparameters and replacing them with an adaptive learning rate, there is an advantage in significantly reducing the need for grid search in model tuning.

**Weaknesses:**

W1. Figure 1 illustrates the distribution of the gradient's standard deviation as training progresses. While it is clear that the expectation in an OLS setting is affected by the standard deviation, the relationship between this formula and dropout remains unclear. For instance, the spiking gradients in Figure 1 could be due to out-of-distribution inputs. However, this does not seem to be directly related to dropout. In my view, to understand any potential connection between Figure 1 and dropout, it would be necessary to show that with dropout set to 0, the gradients remain stable, exhibiting low standard deviation without any spikes.

W2. The authors highlighted instability in LoRA-based fine-tuning methods with fewer epochs. However, through empirical experiments, it was not observed that training had actually stabilized or that the convergence speed had improved.

W3. The experiments lack baseline comparisons. For Table 1, experiments should include basic baselines, such as LoRA, and reference studies [1,2].

W4. (Minor) The ablation results in Figure 4 are not entirely clear. Personally, I believe that instead of using four separate plots to show relative improvement rates for the same data and parameters, it would be more effective to present this information in a single table.

>[1] Zhang, Qingru, et al. "AdaLoRA: Adaptive budget allocation for parameter-efficient fine-tuning." *arXiv preprint arXiv:2303.10512* (2023).

>[2] Jiang, Ting, et al. "MoRA: High-Rank Updating for Parameter-Efficient Fine-Tuning." *arXiv preprint arXiv:2405.12130* (2024).

**Questions:**

Q1. It seems that the size of the adapter needs to be calculated for each batch to determine the adaptive learning rate. How much additional computation does this require? Specifically, how much does it increase the actual runtime compared to LoRA?

Q2. Equation 3 assumes a multi-linear model. However, in most large language models (LLMs), several activations, FFNs, and normalizations are applied after the LoRA adapter. In such cases, the formula in line 279 may not hold. For instance, if normalization is applied after the layer, $\Vert f_L(x) \Vert$ could be smaller than the calculated value. This raises the question of whether the Ripple Effect could still occur in LLMs under these circumstances.

Q3. Figure 3 illustrates performance improvements through figures for comparison with the baseline. However, in my opinion, given the relatively modest performance gains, this does not effectively demonstrate the model's superiority. Could you present the performance more clearly by indicating the absolute values?

---

### Official Review · Reviewer_ehUP · 2024-11-04

**Soundness:** 2
**Presentation:** 2
**Contribution:** 2
**Rating:** 5
**Confidence:** 3

**Summary:**

This paper first identifies three main challenges in fine-tuning with LoRA: dropout, zero initialization, and the difficulty of setting an appropriate scaling factor. The authors propose solutions to these issues by adaptively setting the learning rate. Through various experiments, they demonstrate that their methods outperform LoRA, DoRA, and other ALLoRA-like baselines.

**Strengths:**

1. The concept of adaptive learning for LoRA is somewhat innovative, although the formulation resembles adaptive optimizers in full fine-tuning, such as AdaFactor.
2. The experiments encompass a diverse range of model types and sizes, addressing both vision and language models.

**Weaknesses:**

The primary issue with this paper lies in its clarity and coherence, particularly in Sections 3.1 and 3.2. The mathematical reasoning presented is often inconsistent with the conclusions drawn.

For instance, in Section 3.2, the claim that “as the training set size (n) decreases, the optimization landscape for V becomes degenerate” is not convincingly supported by the preceding discussion on the Hessian product.

In page 4, it is unclear to me how the formula (the "how far off" expectation bound) helps to argue the main conclusion of this section. The paragraph after the formula just does not make sense to me.

Additionally, there is a notable lack of comparison with relevant prior work that addresses similar challenges outlined in Sections 2 and 3. Specifically, regarding initialization, existing studies such as PiSSA[1], MiLoRA[2], and LoRA-GA[3] have already highlighted the benefits of non-zero initialization for LoRA. The discussion on scaling factor selection has also been explored by RSLoRA[4].

Furthermore, while the authors cite LoRA+[5] in relation to learning rate adjustments, there is no direct comparison between the fixed learning rate strategy of LoRA+ and the adaptive approach of ALLoRA. A thorough comparison with these established methods is necessary to substantiate ALLoRA’s claims and enhance the paper’s credibility. I agree that some of the mentioned works are very recent and may be considered concurrent work considering ICLR policy. However, given the pace of the development of this direction, and many of these recent papers dealing with very similar issues and adopting similar ideas (some from arguably more principled way), it is difficult to assess the novelty and contribution of the present submission without comparison to any of them. Even if I ignore all these recent works, I am not super exicited with the theoretical reasoning and empirical results of the paper. Hence, I think the submission may not pass the bar of ICLR acceptance.

[1]	 Meng, Fanxu, Zhaohui Wang, and Muhan Zhang. "Pissa: Principal singular values and singular vectors adaptation of large language models." arXiv preprint arXiv:2404.02948 (2024).
[2]	 Wang, Hanqing, et al. "MiLoRA: Harnessing Minor Singular Components for Parameter-Efficient LLM Finetuning." arXiv preprint arXiv:2406.09044 (2024).
[3]	 Wang, Shaowen, Linxi Yu, and Jian Li. "LoRA-GA: Low-Rank Adaptation with Gradient Approximation." arXiv preprint arXiv:2407.05000 (2024).
[4]	Kalajdzievski, Damjan. "A rank stabilization scaling factor for fine-tuning with lora." arXiv preprint arXiv:2312.03732 (2023).
[5]	 Hayou, Soufiane, Nikhil Ghosh, and Bin Yu. "Lora+: Efficient low rank adaptation of large models." arXiv preprint arXiv:2402.12354 (2024).

**Questions:**

1. In Figure 3, the authors compare LoRA, ALLoRA-0 (ALLoRA without dropout), and ALLoRA. From the previous sections, it is apparent that dropout is detrimental to LoRA’s performance given the authors only fine-tune for two epochs. I am curious about the comparison between ALLoRA-0 and LoRA without dropout.
2. In Section 3.1, why does the average variance of the gradient decrease slightly during training while the worst-case scenario increases? Does this observation hold for other models, including LLMs?
3. In page 2, the authors called the three issues of LoRA "fatal flaws". I think the wording here is not appropriate. Vanilla LoRA has been also used extensively in many applications and it worked just fine.

---

### Meta-Review · Area_Chair_AHHT · 2024-12-16

**Metareview:**

This work first analyzed the limitations of the vanilla LoRA. In total, the vanilla LoRA has three weak points, some of which were already known, e.g., the scaling effect of LoRA. In addition, reviewers pointed out (potentially) confusing parts in their proofs and analyses. However, the authors did not answer appropriately. I consider that their proofs and analyses are not correct after seeing the silence of the authors during the rebuttal phase. As reviewers said, this work lacks theoretical analyses and comparison with existing work. There have been recently proposed many LoRA enhancements and the authors need to consider them seriously before resubmission.

**Additional Comments On Reviewer Discussion:**

The authors did not participate in the rebuttal discussion.

---

### Decision · Program_Chairs · 2025-01-22

Reject